# Rural Housing Rental Rates in China: Regional Differences, Influencing Factors, and Policy Implications

**Li Huang [1,2,\*], Minjie Zheng [3] and Rongyu Wang [4]**

1   School of Public Administration, China University of Geosciences (CUG), Wuhan 430074, China
2   Key Laboratory for Research on Rule of Law, Ministry of Natural Resources, Wuhan 430074, China
3   Business School, The University of Edinburgh, Edinburgh EH8 9JS, UK; s2148607@ed.ac.uk
4   School of Public Affairs, Xiamen University, Xiamen 361005, China; wangrongyu@xmu.edu.cn
\*   Correspondence: huangli@cug.edu.cn

**Abstract:** Through recognition and mastery of the regional differences and influencing factors of China's rural housing rental rates, we can better understand changes in the functional attributes of homesteads and deepen the reform of "separating rural land ownership rights, contract rights, and management rights" of homesteads. Accordingly, this paper uses village residence data from the China Labor-force Dynamics Survey to measure the degree of regional differences in rural housing rental rates at the province level and empirically analyze the influencing factors with villages (residences) as measuring unit. The study yields four main findings. First, rural housing rental behavior exists to varying degrees in the vast majority of provinces nationwide. Second, according to the spatial distribution pattern, rural housing rental rates are generally high in the eastern coastal region and low in the central, western, and northeastern regions, mainly reflecting unique characteristics of the eastern region. Third, although the level of economic development is important, it is not the only factor explaining regional differences in rural housing rental rates. Fourth, rural housing rental rates are mainly influenced by a combination of three types of factors: physiographic, socioeconomic, and village governance factors. Among them, factors such as proximity to suburban areas, the proportion of non-local permanent residents, annual per capita income, and village infrastructure conditions have significant positive effects, whereas factors such as distance from administrative centers, reliance on funding from the higher-level authority of the village committee, and the degree of harmony between villages and cadres have significant negative effects. By interpreting the policy implications of these findings, we hope to provide a reference for localized, categorical reform of the homestead system.

**Keywords:** homestead; rural housing; rental rate; regional differences; influencing factors; policy implications

## 1. Introduction

For decades, the rural residential land system in China featured administrative allocation and rigorous land use regulation. Specifically, rural residential land resources were allocated by administrative command following the principle of egalitarianism, and rural households obtained rural residential land free of charge by virtue of collective membership [1]. However, each rural household was only eligible to possess one plot of rural residential land, the area of which should not exceed the officially set standard [2]. Moreover, the transfer of rural residential land between rural households and urban land users was strictly prohibited, whereas the transfer within rural households in the same village was only conditionally permitted [3]. In practice, rural households tended to occupy excessive plots and areas of rural residential land due to unpaid land use, a lack of village-level land use planning, and weak monitoring efforts [4]. Rapid industrialization and urbanization induced mounting rural-to-urban immigration, leaving a large amount of

underused and idle rural residential land [5]. Furthermore, rural households were unable to gain property revenue and increase their income by transferring rural residential land due to the aforementioned regulation [6].

In this regard, from 2015 to 2019, the Ministry of Land and Resources[1], empowered by the National People's Congress, selected 33 county-level pilot areas in the eastern, central, and western regions of China for rural land reform that took rural residential land as one of its key targets [7]. The rural residential land reform aims at increasing land use efficiency and increasing rural income through leveraging market mechanisms and relieving administrative regulation. Since 2020, a new round of reforms for the rural residential land system has been launched in China, seeking stable progress and maintaining sufficient time-tested patience to establish a new institutional framework [8]. Particularly, rural residential land rentals are always conceived as a major reform task. In practice, however, rural residential land rentals show a significant regional divergence in China. In certain areas, the land rental market is active, and the rental rate strikes a fairly high point, while in other areas, the "thin market" [9] occurs with a relatively low rental rate. Thus, why does the above regional difference in rural residential land rental rates emerge?

A large body of literature has explored the determinants of the supply-demand relation in the rural residential land rental market worldwide. Most studies indicate that location exerts a profound impact on rural residential land rental rates [10]. Economic conditions are recognized as another salient factor [11,12]. However, given the collectively-owned land property rights regime in rural China, diverse functional attributes pertained to rural residential land deserve further investigation to capture the underlying patterns of the observed regional differences across the country. Residential security remains the main functional attribute of rural residential land. Under the current "half-work, half-farming" livelihood model, based on the intergenerational division of labor in rural China, young adults move to the city while the middle-aged and older people still rely on rural residential land as means of production and living, providing financial support for their children to settle in the city [13]. More importantly, as farmers usually struggle to settle in urban areas, a viable option is required to enable their return to their hometowns, thus avoiding the phenomenon of shanty towns and the resulting social unrest that is common in large developing countries [14]. During the ongoing rural land marketization, the asset attribute of rural residential land is gradually manifesting. Especially in the new stage of high-quality development in China, rural residential land is not only an essential link to urban–rural economic cycle [15] but also an indispensable channel to drive the free flow of various production factors [16]. In the inner suburbs of cities or other areas with superior resource endowments, the redevelopment of underused and idle rural residential land can realize the optimal combination of production factors such as labor, land, industry, and finance [17], increase farmers' property income and boost both rural revitalization and urban–rural integration [18]. Moreover, in the outer suburbs of cities or other areas with large rural population outflow, the paid transfer of rural residential land use rights can provide capital accumulation enabling farmers to settle in urban areas [19]. To the best of our knowledge, the extant literature seldom illustrates the regional differences in rural residential land rental rates in the collectively-owned institutional setting of rural China, especially considering both residential security and asset attributes of rural residential land.

Therefore, this study intends to answer the following question: why do rural residential land rental rates differ in regions of China, given its residential security and asset attributes? To that end, a theoretical framework consisting of physiographic, socioeconomic, and village governance factors is developed to conceptualize the impact of functional attributes of rural residential land exerting on rural residential land rental rates across China. A comprehensive method of Dagum Gini coefficient, Logit model, and Tobit model is adopted to corroborate the theoretical inferences, using village residence data from the China Labor-force Dynamics Survey (CLDS) conducted by the Social Science Research Center of Sun Yat-sen University. This study probably contributes to the international discussion on the supply-demand relation in the rural residential land rental market by

highlighting the joint effects of various functional attributes with the empirical evidence in rural China. Furthermore, this study shed light on improving the governance effectiveness of the rural residential land rental market in China and other countries also embedded in collectively-owned land property rights regimes.

The rest of the paper is structured as follows. The second section details the institutional background of the study. The third section details the methods and data used in the study. The fourth section elaborates on the findings of empirical analysis, underscoring the effects of physiographic, socioeconomic, and village governance factors. The last section concludes with policy implications.

## 2. Rural Homestead System and Rural Housing Rental Behavior in China: A Brief Overview of the Institutional Background

### 2.1. The Birth and Evolution of the Planned Allocation System of Rural Homesteads

After its founding, New China set out to pursue the priority development of heavy industry and to implement the strategy of catch-up development. Therefore, rural areas were needed to provide capital, labor, and raw materials for the primitive accumulation of industrialization [20], realized through the "industrial occupation of agricultural profits." In 1958, featuring "large in size and collective in nature," a system of rural people's communes was established, uniting government administration with commune management. Under this system, the government directly controlled the operation of the rural economy through a near-absolutely even distribution system, a highly centralized labor and production management system, a complete monopoly on the unified purchase and sale of agricultural and sideline products, and intensive control over commune members' migration and choice of jobs [21]. Consequently, the commodity properties of rural factors of production, such as land, capital, and labor, were completely eliminated, and the government at all levels managed and regulated the various factors of production through directive and guiding plans.

In this historical context, rural housing land—which carried the dream of "homeowner ship"—was also subsumed into the basic framework of the planned economy during the People's Commune Movement. From 1949 to 1952, the period of land reform in New China, farmers enjoyed full ownership of land, and almost all farming households nationwide received land and property ownership certificates issued for household units. Homesteads, as part of rural private land, were equally protected by law. This framework of farmers' ownership of houses and homesteads continued during the subsequent period of agricultural cooperation from 1952 to 1958 [22]. It was not until the People's Commune period (1958–1983) that the private right to homesteads was fully supplanted by collective ownership of rural land. Issued in 1962, the *Regulations on the Work of the Rural People's Commune (Revised Draft)* mandated that all homesteads of commune members be returned to the commune, and that "all homesteads of commune members are not allowed to be rented or traded." In 1963, the Central Committee of the Communist Party of China issued the *Notice on Some Supplementary Provisions on the Issue of Social Homesteads for Members of the Communist Party of China.* The notice contained more detailed provisions on homestead issues, stipulating that Communist Party members had the right to use but not own homesteads, as well as detailing the ways of applying for and acquiring homesteads on demand and without compensation. Thus, for farmers, the homestead ceased to be an important means of livelihood and only retained the attribute of residential security, with planned management of the homestead used to guarantee farmers' residential rights and the basic stability of the countryside[2]. There were two important features of the management system of rural homesteads under the planned economy. First, it emphasizes the "planning" function of the state, whereby the state and the government allocate and distribute the means of production and living. Specifically, in the case of homesteads, the government set specific rules and rural collective economic organizations were responsible for implementing them. Second, "egalitarianism" became the guiding ideology for homestead allocation, eliminating the possibility of farmers profiting from

homesteads and ignoring the different needs of individuals but ensuring that every farmer has a place to live [23].

After China's reform and opening up, the planned economy was gradually replaced by the market economy, and the single public ownership system progressed toward multiple ownership systems. However, the planned allocation system of homesteads has essentially persisted. Although a series of laws and regulations have been introduced to strengthen the management of homesteads, and a more comprehensive management structure has been established, relevant policies aim to curb the trend of non-agriculturalization of arable land and the expansion of homestead areas [24,25]. Even the *Property Law* promulgated in 2007 did not alter the planned allocation system of homesteads. Indeed, Article 153 stipulates that "the acquisition, exercise and transfer of the right to use homesteads shall be governed by the Land Management Law and other laws and relevant state regulations."

In summary, China's current system of homestead allocation and management was born in the planned economy period and has very strong characteristics of planned distribution: the law prohibits the individual transfer of the right to use homesteads; this right can only be acquired by applying for collective membership, which is subject to various legal constraints; before an administrative permit is issued, the village (township) government must examine and approve the application—which somewhat negates the ownership rights of rural collective economic organizations. Furthermore, homesteads can only be used as rural villagers' self-built houses, and still cannot be transferred outside the collective economic organization. The above arrangements of the homestead management system all aim at giving farmers basic housing security, thus achieving overall stability in rural society and ensuring that each farmer can fairly obtain a place to live in peace and security, in circumstances of scarce land resources.

### 2.2. Changes in the Functional Attributes of Rural Homesteads and the Proliferation of Rural Housing Rental Behavior

After the reform and opening-up, China's increasingly affluent farmers commonly pursued the aims of expanding their living areas and improving living conditions. Their enthusiasm for building houses manifested across the country, resulting in a counter-trend development pattern of decreasing rural resident population and rising rural housing area per capita. Qu and Zhu noted that this counterintuitive development pattern may be related to the changing functional attributes of homesteads. By conducting in-depth interviews and a questionnaire survey in three typical villages in Changsha (respectively representing urban, inner suburban, and outer suburban villages), they discovered that the residential security attribute of the homestead is weakening while the asset attribute is increasingly resurgent. In outer suburban villages, with a large number of surplus rural laborers moving to urban areas for work and business, a large number of homesteads have become idle. Meanwhile, in urban and inner suburban villages, the social security system offers significantly better coverage than in outer suburban villages, and some farmers have purchased urban commercial housing. It appears that rapid urbanization and increasingly frequent population movement between urban and rural areas have fostered the potential asset attribute of homesteads in urban and inner suburban villages. This trend will likely spread to homesteads in outer suburban villages as economic and social development continues [26].

The gradually emerged asset attribute of homesteads is particularly evident in the rental behavior of rural housing. Although the law prohibits farmers from renting out their homesteads, there is a general trend of farmers using their homesteads to build houses to rent in areas with superior location conditions and infrastructure facilities. Fang and Tian conducted a survey of villagers from 24 villages in Jianggan, Gongshu, Yuhang, and Binjiang districts of Hangzhou, located at the intersection of urban and rural areas. Their research revealed a relatively active rural housing rental market in the surveyed areas, with 62.8% of farmers renting out houses, an average total rental area of 163.33 m$^2$, and an average annual rental income of 29,900 yuan [27]. In Yin and Cai's random-sample survey of

411 households in eight urban villages in Wuhan, 14.4% of households engaged in housing rental behavior before demolition, and the average housing rental income per household was 4764 yuan/year [28]. Xuan's study of informal housing rental in villages and towns in the Pearl River Delta—the "factory of the world"—revealed that agricultural housing rental has effectively addressed the rental needs of tens of millions of migrant workers, while also compensating for the shortcomings of the formal housing system [29]. In the national reform of the three rural land systems launched in 2016, several pilot counties, such as Yiwu, Meitan, and Luxian, have introduced relevant policies to regulate the rural housing rental phenomenon and promote the growth of farmers' property income [30].

Nevertheless, with limited available data from studies using a large sample size and offering high credibility and wide coverage, both sides of the debate introduced earlier can find evidence from different regions and time periods to support their arguments on many basic questions, such as whether the rural housing rental phenomenon exists only in urban and suburban villages of the developed eastern coastal region. To bridge the differences in opinion and seek consensus on reform, this paper uses data from the CLDS database to empirically analyze the regional differences in rural housing rental behavior across the country and what factors influencing it. The study should provide more precise evidence to inform the governance and classification of homesteads.

## 3. Data sources and Research Methods

### 3.1. Data Sources

The data for this study were obtained from the CLDS database, operated by the Social Science Research Center of Sun Yat-sen University. The CLDS is a continuous village/household survey conducted every two years with a rotating sample method, which better reflects China's rapid changes and combines the characteristics of cross-sectional and tracking surveys. The database offers significant advantages in three respects. First, it provides a wide survey scope and a large sample size. The CLDS sample covers 29 provinces (cities and autonomous regions) in China[3], with over 300 villages, 14,000 households, and 20,000 labor-force individuals included in each survey round. Second, the CLDS has good sample representativeness. It uses a multi-stage, multi-level probability sampling method proportional to the size of the workforce, thereby ensuring that the sample is well representative of the national population. Third, it offers rich research content. The CLDS includes tracking databases at three levels: labor force, households, and village residences. Most relevant for this study's purposes, the village database includes the following: socioeconomic overview, land and housing situation, population and structure, grassroots organizations, democratic elections, public affairs participation, environment and infrastructure, grassroots governance, and history. The CLDS thus provides high-quality basic data for studying regional differences in rural housing rental rates and their influencing factors.

This paper adopts data from the 2012 national baseline survey (CLDS2012), the first follow-up survey (CLDS2014), and the second follow-up survey (CLDS2016). The overall study interval is 2012–2016[4], during which 623 completed questionnaires were obtained from across 29 provinces (municipalities and autonomous regions).

### 3.2. Variable Descriptions

#### 3.2.1. Dependent Variable

This study's dependent variable is the "rural housing rental rate." In the analysis of regional differences in the rural housing rental rate, this variable is calculated at the province level as the proportion of surveyed households in the province with housing rental behavior. In the analysis of factors influencing the rural housing rental rate, the dependent variable is calculated at the village level as the proportion of total households in the village that have rented out housing.

### 3.2.2. Independent Variable

Many factors influence whether farmers rent out their own houses. By drawing on the existing literature most relevant to our research objectives, we select physiographic factors, socioeconomic factors, and village governance factors as independent variables.

Physiographic Factors

Jia et al. clearly point out that physical geography is a key factor in the conversion of homesteads from resources to assets [31]. By quantifying the multifunctional value of homesteads, Yuan et al. found that homesteads adjacent to main village roads and close to urban areas are more efficiently utilized, whereas those far from main roads are prone to be idle [32]. In terms of macro-geographical distribution, the non-agricultural value and asset function of the homestead are highlighted in plain areas because of the large population and concentration of settlements, while the residential security function of the homestead is relatively higher in remote mountainous areas where settlements are scattered. Qi et al. found that physiographic conditions are stable in the long term and that a superior physiographic endowment can play a fundamental role in supporting the transformation of homestead function. This study measures physiographic factors using three independent variables: whether or not the village is located on the outskirts of a city, the distance of the village from the nearest county/district government, and the distance of the village from the nearest township government/sub-district [33].

Socioeconomic Factors

In a case study of homestead system reform in Luxian County (Sichuan Province), Liu and Xiong argue that the economic structural changes and village transformation caused by continuous industrialization and urbanization are the inherent needs of homestead system reform [34]. Related quantitative studies have also confirmed that change in the functional attributes of homesteads is influenced by various socioeconomic factors, such as the economic development level [35], per capita infrastructure investment [36], rural labor-force transfer [37], and village industrial structure [38]. This study measures socioeconomic factors using four independent variables: the proportion of non-local permanent residents, the proportion of residents working and doing business outside the village, the annual per capita income, and the village's infrastructure conditions. Table 1 details how each variable is measured, including the units (where applicable).

Village Governance Factors

Homesteads serve as a typical closed public pond resource. The village collective is not only the main body responsible for formulating and implementing homestead management policy but also the specific field for policy target groups to carry out conceptual collision, interest game, and action interaction [39]. Therefore, homestead land rights are embedded in the village governance field [40], and the influence of village governance factors on the change in homestead functional attributes must be taken into account. This study measures village governance factors using three independent variables: the number of village representative assemblies/village assemblies held in the village, reliance on funding from the higher-level authority of the village committee, and the degree of harmony between villagers and village committee cadres. As above, Table 1 details how each variable is measured, including the units (if applicable).

**Table 1.** Definitions and descriptive statistics of the study variables.

| Category | Variable Description | Definition | Unit | Mean | Median | Standard Deviation |
|---|---|---|---|---|---|---|
| Dependent variable | Is there any external rental housing in the village | Yes = 1; No = 0 | — | 0.05 | 0 | 0.13 |
| | Housing rental ratio in the village | Number of households with housing for rent in the village/Total number of households in the village | % | 0.04 | 1.15 | 7.32 |
| Physiographic factors | Is the village located on the outskirts of a city? | Yes = 1; No = 0 | — | 0.11 | 0 | 0.31 |
| | Distance of the village from the nearest county/district government | Distance of the village from the nearest county/township government | km | 25.69 | 20 | 21.56 |
| | Distance of the village from the nearest township government/sub-district | Distance of the village from the nearest township government/sub-district | km | 5.68 | 4 | 5.62 |
| Socioeconomic factors | Proportion of non-local permanent residents | Number of non-local migrant population/Total population of the village | % | 0.49 | 0.01 | 9.14 |
| | Proportion of residents working and doing business outside the village | Number of residents working and doing business outside the village/Total population of the village | % | 0.20 | 0.07 | 0.26 |
| | Annual per capita income | Annual per capita income of household registration population, with CPI data for each province uniformly converted to 2012 constant price | 10,000 yuan/year | 0.94 | 0.6 | 1.39 |
| | Village infrastructure conditions | Proportion of hardened roads for traffic to all roads in the village | % | 58.84 | 60 | 29.58 |
| Village governance factors | Number of village representative assemblies/village assemblies held in the village | Number of village representative assemblies/village assemblies held in the village during the year | — | 4.93 | 3 | 6.55 |
| | Reliance on funding from the higher-level authority of the village committee | Office expenses paid by higher-level authority/Total office expenses of the village | % | 0.63 | 0.8 | 0.42 |
| | Degree of harmony between villagers and village committee cadres | Relatively low = 1; average = 2; relatively high = 3; very high = 4 | — | 3.00 | 3 | 0.71 |

*3.3. Research Methodology*

3.3.1. Dagum Gini Coefficient and Its Decomposition by Subgroups

Compared with the traditional Gini coefficient and the Theil index method, the Dagum Gini coefficient and its decomposition by subgroups better solve the problems of crossover between samples, the distribution status, and net regional differences of sub-samples as well as more precisely identify the sources of regional differences. Referring to Dagum, Ogwang, and other related prior research [41,42], this study divides the national administrative regions into $M$ parts, with $f_{ih}$ and $f_{jr}$ as the rural housing rental rate of a province or municipality in region $i$ $(j)$. $\mu$ denotes the mean value of the national rural housing rental rate, $n$ is the number of all provinces and municipalities, while $n_i$ and $n_j$ represent the

number of provinces and municipalities in region $i$ ($j$). Thus the Gini coefficient is obtained using Equation (1):

$$G = \frac{1}{2n^2\mu}\sum_{i=1}^{M}\sum_{j=1}^{M}\sum_{h=1}^{n_i}\sum_{r=1}^{n_j}|f_{ih} - f_{jr}| \tag{1}$$

The Gini coefficient thus calculated can be further decomposed into three components: the contribution of within-group (intra-regional) variation ($G_\theta$), the net contribution of between-group (inter-regional) variation ($G_{nb}$), and the contribution of between-group superdensity ($G_\gamma$). The relationship among these three components is as shown in Equation (2):

$$\begin{aligned} G &= G_\theta + G_{nb} + G_\gamma \\ &= \sum_{i=1}^{M} m_i s_i G_{ii} + \sum_{i=2}^{M}\sum_{j=1}^{i-1}(m_j s_i + m_i s_j)G_{ij}H_{ij} + \sum_{i=2}^{M}\sum_{j=1}^{i-1}(m_j s_i + m_i s_j)G_{ij}(1 - H_{ij}) \end{aligned} \tag{2}$$

### 3.3.2. Logit Model

In Section 5: analysis of factors influencing the rural housing rental rate, this study empirically analyzes the possible factors influencing the rural housing rental rate with village residents as the unit of analysis. For a village, the first issue to investigate is whether there is any housing rental behavior: we define $Y = 1$ to mean that there is housing rental behavior in the village and $Y = 0$ to mean that there is no such behavior. For regression analysis of a dichotomous discrete variable, a binary logit model is appropriate. Thus, we develop the regression model shown in Equation (3):

$$Prob(Y_i) = P_i = \frac{e^{\beta_0 + \beta_j X_{ij}}}{1 + e^{\beta_0 + \beta_j X_{ij}}} = \frac{1}{1 + e^{-(\beta_0 + \beta_j X_{ij})}} \tag{3}$$

where $Y_i$ denotes the probability of housing rental behavior in the $i$th village residence, $i$ is the village residence number, $X_{ij}$ denotes the $j$th influencing factor of the $i$th village residence, $\beta_j$ denotes the coefficient value of the $j$th influencing factor, and $j$ is the influencing factor number. After model transformation, a binary logit model for analyzing the rural housing rental rate is obtained, as shown in Equation (4):

$$\ln\frac{P_i}{1 - P_i} = \alpha + \sum_{j=1}^{n}\beta_j L_{ij} + \varepsilon_i \tag{4}$$

where $\frac{P_i}{1-P_i}$ represents the ratio of the presence of housing rental behavior to the absence of housing rental behavior, and $\varepsilon_i$ is the residual term.

### 3.3.3. Tobit Model

Among villages where households engage in rural housing rental behavior, there are some differences in the degree of development of the housing rental market. Accordingly, this study introduces the village housing rental ratio to characterize the development of the rural housing rental market. As shown in Table 1, the value of this ratio ranges from 0 to 1. As a restricted dependent variable, the village housing rental ratio is suitable for regression analysis using a Tobit model. The Tobit regression model developed for this purpose is shown in Equation (5):

$$Z_i = \gamma_0 + \gamma_j X_{ij} + \mu_i \tag{5}$$

where $Z_i$ is the housing rental ratio of the $i$th village, $i$ is the village residence number, $X_{ij}$ denotes the $j$th influencing factor of the $i$th village residence, $\gamma_j$ denotes the coefficient value of the $j$th influencing factor, $j$ is the influencing factor number, and $\mu_i$ is the residual term.

## 4. Analysis of Regional Differences in the Rural Housing Rental Rate in China

### 4.1. Overall Spatial Distribution Pattern

As the CLDS adopts a rotating sample method, different villages and residences are included in each round of the survey. Therefore, when investigating the rural housing rental rate at the province level, we can only analyze regional differences in the spatial dimension and cannot analyze evolution in the temporal dimension. Figure 1 depicts the regional differences[5] in rural housing rental rates.

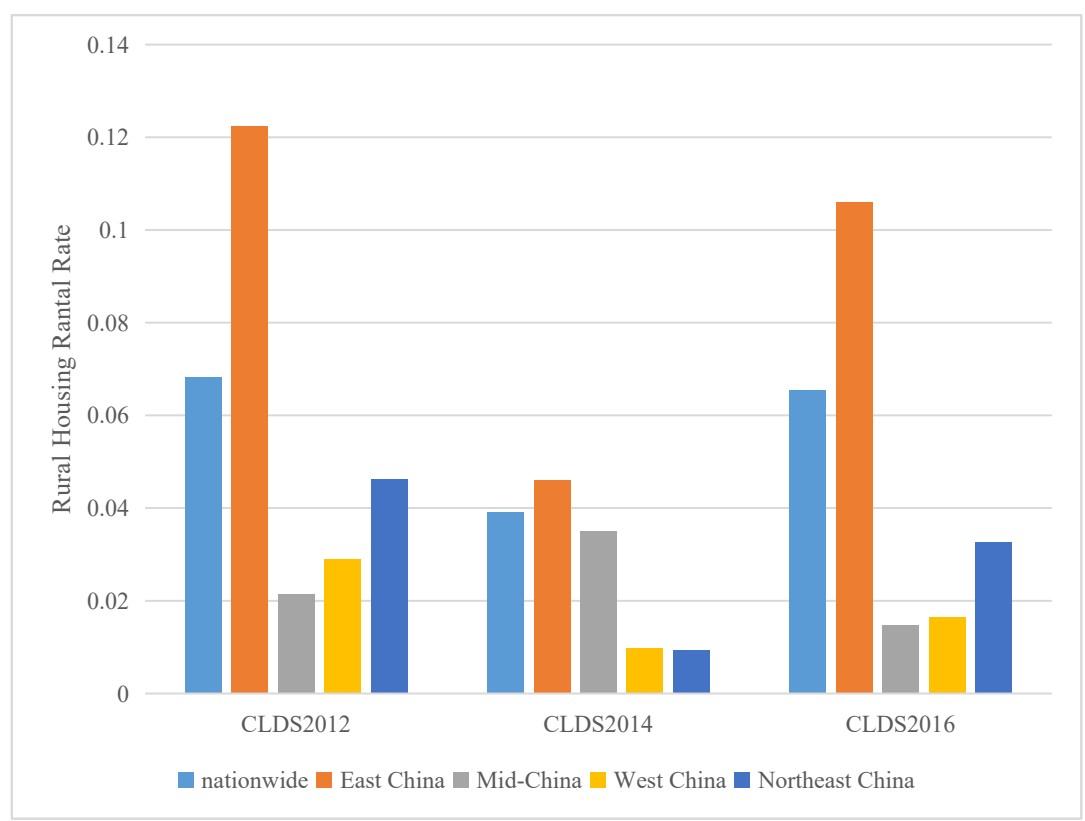

**Figure 1.** General characteristics of the rural housing rental rate.

In the three rounds of the CLDS, the average rural housing rental rate in China is 5.76%[6], indicating that a certain number of farm households have rented out their houses nationwide and that the asset attribute of homesteads has manifested to a certain extent. From a regional perspective, the average rural housing rental rate in eastern provinces and cities is 9.14%, much higher than in the northeastern region (2.94%), central region (2.37%), and western region (1.85%). We can tentatively judge from these findings that there are significant regional differences in the rural housing rental rate. The current problem of unbalanced and insufficient development in China is also reflected in the development of the rural housing rental market and the changing functional attributes of homesteads.

### 4.2. Provincial Differences in the Rural Housing Rental Rate

To further investigate province-level differences in the rural housing rental rate, this study divides provinces and municipalities into four tiers (as shown in Table 2) and analyzes them separately for comparison.

**Table 2.** Basic situation of China's rural housing rental rates in each province.

| | Province-Level Rural Housing Rental Rate (%) | CLDS2012 | CLDS2014 | CLDS2016 |
|---|---|---|---|---|
| Tier 1 | ≥10% | Beijing, Shanxi, Zhejiang, Guangdong | Beijing, Tianjin, Shanxi Zhejiang | Tianjin, Zhejiang |
| Tier 2 | 5–10% | Tianjin, Liaoning, Fujian, Yunnan, Shanxi | Guangdong | Beijing, Hebei, Shanxi Fujian, Guangdong, Guizhou |
| Tier 3 | 1–5% | Neimenggu, Anhui, Henan, Hubei, Ningxia | Henan, Liaoning, Jilin, Jiangsu, Anhui, Fujian, Hubei, Chongqing, Shanxi | Liaoning, Jiangsu, Henan, Sichuan, Shanxi, Gansu, Ningxia |
| Tier 4 | 0–1% | Hebei, Jilin, Heilongjiang, Jiangsu, Jiangxi, Shandong, Hunan, Guangxi, Chongqing, Sichuan, Guizhou, Gansu | Neimenggu, Heilongjiang, Jiangxi, Shandong, Henan, Hunan, Guangxi, Sichuan, Guizhou, Yunnan, Gansu, Ningxia, Xinjiang | Neimenggu, Jilin, Heilongjiang, Anhui, Jiangxi, Shandong, Hubei, Hunan, Guangxi, Chongqing, Yunnan, Xinjiang |

As shown in Table 2, the provinces and cities in Tier 1 (rural housing rental rate ≥ 10%) are basically those in the developed eastern coastal region, except for Shanxi Province. Notably, Zhejiang Province is in Tier 1 in all three rounds of the CLDS. Published on 10 June 2021, the *Opinions of the CPC Central Committee and the State Council on Supporting Zhejiang's High-Quality Development and Building a Common Wealth Demonstration Zone* propose to:

> smooth the economic cycle in urban and rural areas, and break down the institutional barriers that restrict the equal exchange and two-way flow of factors in urban and rural areas . . . [and to] broaden the channels of property income of urban and rural residents, and explore the right to use and gain the right to increase the factor income of middle- and low-income groups through the land, capital, and other factors.

These and other reform initiatives reflect the central government's expectation for Zhejiang Province to fully mobilize the role of land elements, including homesteads, in social wealth creation, and to increase the property income of all residents through multiple channels. Provinces and cities in Tier 2 (5% ≤ rural housing rental rate < 10%) are mostly in the eastern coastal region but some are in the central and western regions, such as Yunnan Province, Shaanxi Province, and Guizhou Province. This suggests that the economic development level is important but not the only factor influencing rural housing rental rate and that the rate may be affected by other factors in different regions. The provinces and cities in Tier 3 (1% ≤ rural housing rental rate < 5%) and Tier 4 (rural housing rental rate < 1%) are primarily in the central and western regions, and some have a rural rental housing rate of zero (e.g., Hunan (CLDS2014), Jilin (CLDS2012 and CLDS2016), and Heilongjiang (CLDS2014 and CLDS2016)). The asset property function of homesteads hardly manifests in these provinces and cities. It is also noteworthy that Jiangsu Province and Shandong Province, which have high levels of economic development, are in Tier 3 or 4 in all three rounds of the CLDS, indicating that their strong economic development is not yet nurturing the asset attribute of homesteads.

To more intuitively reflect spatial divergence in the rural housing rental rate across China, Figure 2 plots the spatial distribution of the rate by province and municipality for each round of the national-level survey. The spatial distribution exhibits three key characteristics.

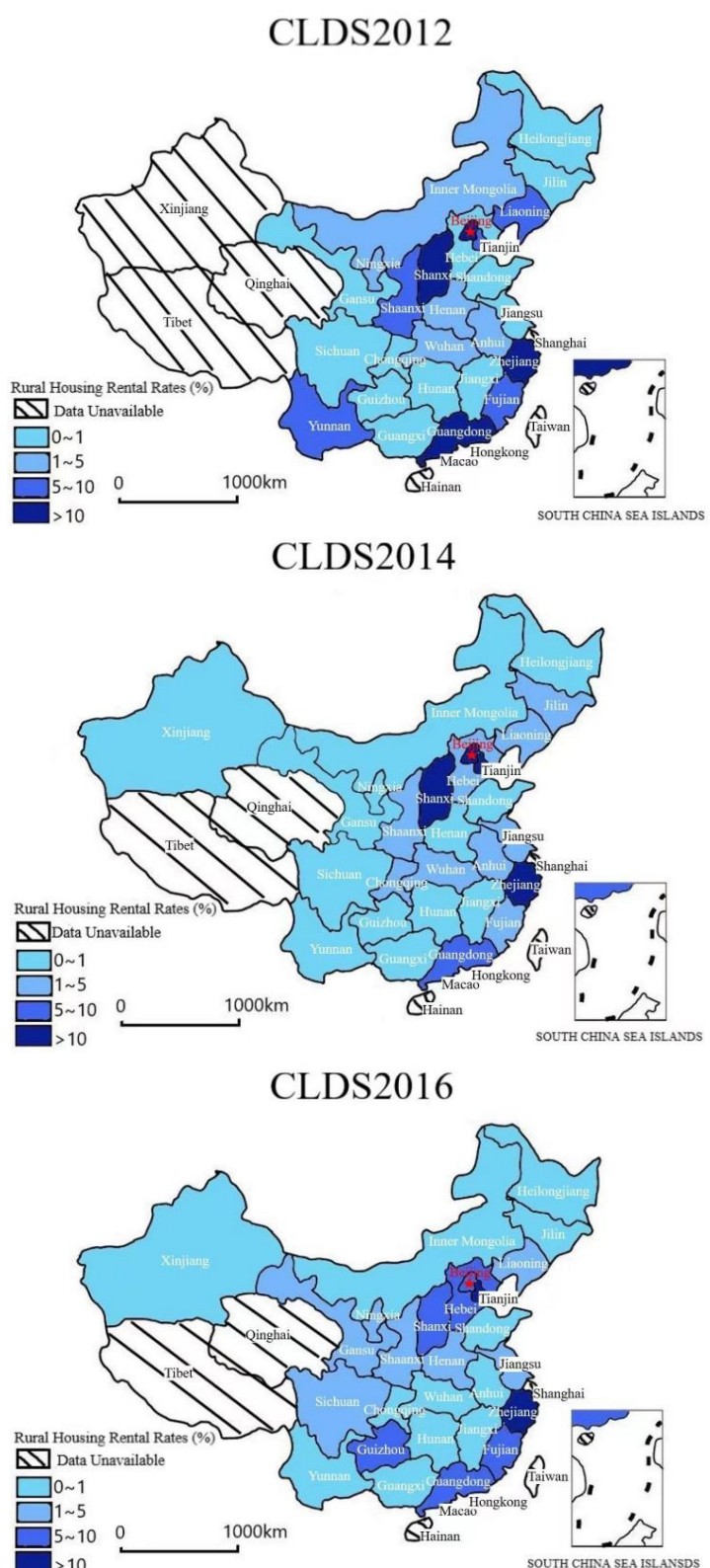

**Figure 2.** Spatial pattern of China's rural housing rental rate.

First, rural housing rental behavior (rural housing rental rate > 0) exists in the vast majority of provinces nationwide, indicating that the basic function of rural housing has, in fact, broken through the limits of the residential security attributes. In not only eastern but

also central and western provinces, rural residents have the opportunity to increase their property income by using the asset attribute of residential property.

Second, the rural housing rental rate is high in the eastern coastal region and low in the central, western, and northeastern regions. For example, the rural housing rental rate in Zhejiang Province exceeds 10% in all three survey rounds, but is zero in several provinces; moreover, at the province level, there are huge differences in the rural housing rental rate.

Third, the economic development level is an important factor but not the only determinant of regional differences in the rural housing rental rate. The rental rate of farmhouses in Jiangsu and Shandong provinces, both with high economic development, is below 5% in all three rounds of the CLDS, whereas the rental rate of farmhouses in Shanxi Province, which has a relatively less developed economy, is above 5%. These findings suggest that factors other than the economic development level influence the rural housing rental rate.

### 4.3. Decomposition of Regional Differences in the Rural Housing Rental Rate

To further investigate the sources of regional differences in the rural housing rental rate, the Gini coefficients of the three rounds of CLDS data were measured using MATLAB software according to Equations (1) and (2) for the four major regions of eastern, central, western, and northeastern China. The results are shown in Table 3.

**Table 3.** Comparison and decomposition of the Gini coefficient for the rural housing rental rate across China.

| | Overall Gini Coeffi-cient | Differences between Regions | | | | | | Intra-Regional Differences | | | | Contribution Rate (%) | | |
|---|---|---|---|---|---|---|---|---|---|---|---|---|---|---|
| | | Central vs. East | West vs. North-east | West vs. Cen-tral | Northeast vs. East | Northeast vs. Central | Northeast vs. West | East | Central | West | Northeast | Within Re-gion | Between Re-gion | Super Vari-able Den-sity |
| CLDS 2012 | 0.720 | 0.747 | 0.796 | 0.669 | 0.817 | 0.687 | 0.709 | 0.600 | 0.582 | 0.691 | 0.640 | 25.00 | 53.87 | 21.13 |
| CLDS 2014 | 0.697 | 0.649 | 0.869 | 0.751 | 0.856 | 0.739 | 0.425 | 0.500 | 0.616 | 0.416 | 0.342 | 20.61 | 69.59 | 9.80 |
| CLDS 2016 | 0.684 | 0.771 | 0.799 | 0.570 | 0.879 | 0.707 | 0.672 | 0.436 | 0.563 | 0.544 | 0.667 | 19.95 | 71.55 | 8.50 |

The overall Gini coefficient of the rural housing rental rate ranges from 0.684 to 0.720 in the three rounds of the CLDS, indicating a wide range of regional differences in the rate at the province level: some provinces and municipalities appear to have an incipient rural housing rental market, whereas others are yet to begin developing this market. In terms of inter-regional differences in the rural housing rental rate, the greatest difference is between the east and northeast, and differences between the east and the central and western regions are also significantly greater than the differences between other regions. These results indicate that differences in the rural housing rental rate among China's four major regions are mainly between the east and the rest of the country. The regions with the largest and smallest intra-regional differences varied across the survey rounds, without obvious regularity. Regarding contributions to the Gini coefficient of the rural housing rental rate, inter-regional differences make the highest contribution in all three rounds of the CLDS, but especially in CLDS2014 and CLDS2016. This indicates that the problem of regional differences in the rural housing rental rate mainly originates from differences among the country's four major regions.

Scholars investigating changes in the functional attributes of the homestead in different provinces and cities with very different situations will inevitably reach widely varying conclusions. Therefore, we must further explore the factors that may explain significant regional differences in the rural housing rental rate, so as to provide a direct reference for reform of the homestead system.

## 5. Analysis of Factors Influencing the Rural Housing Rental Rate

Table 4 presents the empirical results of the regression analysis of Equations (4) and (5). Columns (1)–(3) give the regression results for the logit model with the dichotomous dependent variable of whether the village has rental housing; Columns (4) and (5) give the regression results for the Tobit model with the village's rural housing rental rate as the dependent variable. Columns (1) and (4) report the regression results using the full sample; Columns (2) and (5) give the results for the eastern provinces; Columns (3) and (6) show the regression results for the central and western provinces combined.

**Table 4.** Regression results of factors influencing the rural housing rental rate in China.

| | Influencing Factor | Dependent Variable: Does the Village Have Rental Housing? | | | Dependent Variable: Village's Rural Housing Rental Rate | | |
|---|---|---|---|---|---|---|---|
| | | Logit Regression Model | | | Tobit Regression Model | | |
| | | Nationwide (1) | Eastern (2) | Central and Western (3) | Nationwide (4) | Eastern (5) | Central and Western (6) |
| Physiographic factors | Is the village located on the outskirts of a city? | 1.037 ** (3.09) | 1.546 *** (3.29) | 0.120 (0.32) | 0.107 ** (3.07) | 0.120 *** (2.59) | 0.083 * (1.77) |
| | Distance of the village from the nearest county/district government | −0.147 * (−1.69) | −0.106 (−0.91) | −0.339 ** (−2.36) | −0.008 (−0.92) | 0.014 (1.01) | −0.027 *** (−2.78) |
| | Distance of the village from the nearest township government/sub-district | −0.340 *** (−4.54) | −0.131 (−0.77) | −0.425 *** (−4.77) | −0.039 *** (−5.07) | −0.066 *** (−3.59) | −0.023 *** (−3.79) |
| Socioeconomic factors | Proportion of non-local permanent residents | 3.592 *** (4.68) | 7.101 *** (3.93) | 1.415 (1.52) | 0.002 * (1.85) | 0.002 (1.60) | 0.120 (1.64) |
| | Proportion of residents working and doing business outside the village | −0.440 (−1.12) | −0.853 (−1.29) | 0.092 (0.18) | −0.022 (−0.48) | 0.013 (0.18) | −0.014 (−0.31) |
| | Annual per capita income | 0.514 *** (4.15) | 0.652 *** (2.96) | 0.293 * (1.87) | 0.081 *** (5.77) | 0.107 *** (4.63) | 0.022 * (1.65) |
| | Village infrastructure conditions | 0.012 *** (3.52) | 0.015 *** (2.58) | 0.011 ** (2.29) | 0.002 *** (3.92) | 0.001 ** (2.26) | 0.012 *** (2.97) |
| Village governance factors | Number of village representative assemblies/village assemblies held in the village | 0.099 (0.84) | −0.065 (−0.43) | 0.314 (1.63) | 0.020 (1.63) | 0.013 (0.81) | 0.023 (1.47) |
| | Reliance on funding from the higher-level authority of the village committee | −0.333 (−1.38) | −0.004 (−0.01) | −0.376 (−1.01) | −0.089 ** (−3.15) | −0.077 * (−1.81) | −0.037 (−1.19) |
| | Degree of harmony between villagers and village committee officials | −0.231 * (−1.66) | −0.768 *** (−3.48) | 0.238 (1.17) | −0.030 * (−1.79) | −0.084 *** (−3.27) | 0.029 * (1.73) |
| N | | 623 | 294 | 329 | 623 | 294 | 329 |

Note: For the logit regression model, z-values are given in parentheses; for the Tobit regression model, *t*-values are given in parentheses; ***, **, and * represent 1%, 5%, and 10% significance levels, respectively.

### 5.1. Physiographic Factors

The variable "Is the village located in the outskirts of a large or medium-sized city?" is significantly positive at the 5% level in Columns (1) and (4); significantly positive at the 1% level and with a substantively higher coefficient value in Columns (2) and (5); and significantly positive at the 10% level and with a lower coefficient value in Column (6). This indicates that at the national level, rural areas on the outskirts of large and medium-sized cities are more likely to have housing rentals and to have a higher rural housing rental rate and that their homesteads have a higher asset attribute. Moreover, the housing rental market is more mature and active in the suburban rural areas of eastern provinces

compared with those of central and western provinces, a finding consistent with the results of prior research on the rural housing rental market [43,44].

The variable "distance of the village from the nearest county/district government" is significantly negative only in Columns (1), (3), and (6), indicating that at the national level, the further the distance from a county/district government, the more likely the occurrence of the housing rental phenomenon. This is because the supply of commercial housing available to rent is more adequate in areas closer to a county/district government and relatively scarce in more distant areas, where people with housing rental needs can only seek to rent from farmers. Notably, the variable's coefficients were significantly negative in the central and western provinces but not in the eastern provinces. One possible explanation is that in the central and western provinces, the government administrative center is a special political resource with a clustering effect, such that infrastructure and public services are spatially allocated around government institutions [45]. Although a more mature housing market thus emerges around government administrative centers, more distant rural areas will have a gap in the supply of commercial housing due to the lack of public resource investment. By contrast, in the highly economically developed and marketized eastern provinces, the spatial location of government administrative centers does not appear to affect the development level of the rural housing rental market.

The variable "distance of the village from the nearest township government/sub-district" is significantly negative in every Column except Column (2). These results indicate that the greater the distance from the nearest township government/sub-district, the more likely the occurrence of the housing rental phenomenon and the higher the probability of housing being rented out. The mechanism of this variable's effect is similar to that of the second physiographic variable, but the spatial location of the township government/sub-district is evidently more important to the development level of the rural housing rental market than the spatial location of the county/district government.

Combining the results for all three physiographic variables, we infer that the rural housing rental market is more developed in the suburban areas of large and medium-sized cities, and the asset attribute function of homesteads is more obvious. Moreover, at greater distances from the county/district government and from the township government/sub-district, housing rental behavior is more likely to occur. The findings suggest there is strong demand for housing rental in rural areas more distant from cities and towns, and that the development of the rural housing rental market can both cover the vast gaps in and complement the functions of the urban housing rental market.

*5.2. Socioeconomic Factors*

The "proportion of non-local permanent residents" is significantly positive in Columns (1), (2), and (4) but insignificant in the other three Columns. The results indicate that at the national level, the higher the migrant proportion of a village's total population, the more likely that village is to have housing rentals, and this phenomenon is more pronounced in eastern provinces. The inflow of the migrant population is evidently an important factor driving change in homestead functions and the development of the rural housing rental market. With the continuous promotion of rural revitalization strategy and the two-way movement of population between urban and rural areas, renting from farmers is satisfying an immediate need of migrant populations in rural areas with no commercial housing.

The "proportion of residents working and doing business outside the village" is not significant in any of the six Columns. This suggests that despite great regional differences in the economic development level of rural areas and the continuous net outflow of the population in some rural areas [46,47], greater understanding is urgently needed of how to activate the rural housing rental market through separating rural land ownership rights, contract rights, and management rights of the homestead, and how to provide a stable property income for those who go out to work and do business.

This indicates that the higher the annual per capita income in rural areas, the more common the rural housing rental phenomenon and the higher the rural housing rental

rate. Rural areas with higher annual per capita income have a more developed non-farm economy, resulting in higher demand for rental housing [48]. Zhang et al.'s study of Taobao villages in Guangzhou, China, revealed that the development of e-commerce has significantly increased local non-farm employment opportunities, enabled local capital accumulation, and continuously attracted the inflow of non-locals to villages [49]. As the per capita income in rural areas continues to rise and a well-off society is fully built, the demand for rural housing rental will continue to increase, thus making the asset attribute of homesteads increasingly prominent.

The variable "village infrastructure conditions" is also significantly positive in all six Columns, indicating that the greater the proportion of hardened rural roads, the more likely the rural housing rental phenomenon and the higher the rural housing rental rate. Through an empirical analysis of the construction of rural roads, Zhang found that improved transportation infrastructure can significantly increase the number of rural households in non-farm employment, and this effect was more significant for low-grade income households [50]. After the COVID-19 pandemic broke out in 2020, the demand for rural rental housing decreased. Shaoxing, Zhejiang Province launched the "Activation Plan of Idle Rural Houses (Version 2.0)" so as to further revitalize idle rural houses and homesteads, thus turning "dead assets" into "living assets". In this "Activation Plan", an important measure lies in combining the "Activation Plan of Idle Rural Houses" with the construction of beautiful countryside through synchronous and uniform planning and overall implementation so that the cleanliness and beauty of the rural environment, as well as the standardization and modernization of rural infrastructure, are taken as the advantages for the development and utilization of idle farmhouses [51]. With the gradual realization of integrated urban–rural infrastructure planning, construction, and management, the increasing non-farm employment opportunities in rural areas will effectively promote the development of the rural housing rental market.

*5.3. Village Governance Factors*

The variable "number of village representative assemblies/village assemblies held in the village" is not significant in any of the six Columns. This number is often used to measure the degree of democratization of village decision-making [52]. Except for in a few pilot counties, there is a general lack of clear, comprehensive, and direct management documents and supporting measures for the rural housing rental market. The extent to which these gaps can be filled depends largely on the governance capacity of village organizations [53]. Yujiang District, Jiangxi Province focused on comprehensively promoting the autonomous management mode of homestead in its rural land system reform to effectively deal with the impacts of the COVID-19 pandemic. Adhering to the principle of democratic recommendation and consultation by the masses, Yujiang District has established 950 villagers' affairs councils, which are "willing to serve people, able to serve people, and manage to serve people". The autonomous management of the homestead has been realized, people's sense of acquisition and happiness has been obviously improved, and a solid foundation for the development of the rural land markets has been laid by giving full play to the role of villagers' affairs councils in organization, coordination, and supervision [54]. Therefore, to promote the reform of separating rural land ownership rights, contract rights, and management rights of homesteads, it is important to enhancing this governance capacity by improving the democratization of village decision-making, thereby providing a transparent, equal, and safe transaction environment for both parties in the rural housing rental market, and thus promoting the market's development.

The "reliance on funding from the higher-level authority of the village committee" is negative in all Columns but only statistically significant in Columns (4) and (5). These results indicate that the lower the proportion of office expenses paid by superiors, the higher the rural housing rental rate in that area, and this trend is more obvious in the eastern provinces. Overreliance of a village on financial allocation from the higher-level authority indicates the shortage of effective means for the development of the village's

collective economy. A poor level of collective economic development at the village level can affect the development of the rural housing rental market through several channels, including the lack of non-agricultural industry development, lack of non-agricultural employment opportunities, and inadequate supply of village public services. Therefore, by promoting the reform of collective economic organizations and the development of the collective economy, China can effectively promote the development of the rural housing rental market and the asset attribute of the homestead.

The "degree of harmony between villagers and village committee cadres" is significantly negative in all Columns (1), (2), (4), and (5). The results indicate that at the national level and in eastern provinces, lower harmony between villagers and village cadres is associated with the more likely occurrence of rural housing rental and a higher rural housing rental rate. Before the launch of the pilots of the three rural land system reforms in 2015, the transfer of the right to use homesteads was strictly restricted by law, and farmers were not allowed to revitalize idle farmhouses and homesteads by renting them out [55]. In this study's sample, except for the very small number of pilot-county respondents, the reported renting behavior should be regarded as non-public gray trading of agricultural housing use rights without formal approval [56]. Such behavior inevitably conflicts with the management responsibilities of village cadres, thus intensifying the conflict between them and the villagers [57]. In the Pearl River Delta region of China, there exists a specific type of community named "village in city" (between villages and cities), which is featured by rural governance and urban community governance. Villages in cities have a large number of assets from the non-agricultural use of village collective land. Social contradictions caused by the way of allocating "land dividends" have become the focus of village-level governance. The governance focus lies in the contradictions between villagers' autonomy, the decision-making of the "two village committees" and national laws and regulations, as well as the resulting frequent petitions and even mass incidents. In the process of analyzing three typical "villages in cities" cases, namely, Yuancun Village, Hecun Village, and Fangcun Village, Huang, and Ding found that the state intervenes in rural society through top-down governance strategies and behaviors, reorganizes village governance structure, and straightens out the benefit-sharing relationship among state, collective, and village community members, which effectively reduces social disputes resulting from land and housing problems, thus realizing effective governance of villages in cities [58]. In this sense, a harmonious relationship between cadres and the masses is an important guarantee for promoting the reform of separating rural land ownership rights, contract rights, and management rights of homesteads.

## 6. Research Findings

To study the changing functional attributes of homesteads, this paper takes the rural housing rental rate as the entry point and uses CLDS village residence data to investigate the nature and driving factors of the distribution pattern of homestead's asset attributes across different regions of China. The following main inferences are drawn from the findings.

First, rural housing rental behavior exists to varying degrees in the vast majority of provinces across the country. This suggests that the basic function of rural housing has, in practice, broken through the limits of the residential security attribute and that opportunities exist for rural residents to increase their property income by exploiting the asset attribute of homesteads.

Second, in terms of spatial distribution, the rural housing rental rate is high in the eastern coastal region and low in the central, western, and northeastern regions. Analysis of the Gini coefficient and its decomposition by subgroups revealed substantial regional differences in the rural housing rental rate, mainly originating from differences between the eastern region and the other three major regions.

Third, the economic development level is an important factor but by no means the only determinant of regional differences in the rural housing rental rate. Provinces and cities with higher levels of economic development do not necessarily show a higher rural

housing rental rate; similarly, some provinces and cities with lower economic development levels exhibit a higher rate.

Fourth, the rural housing rental rate is mainly influenced by a combination of three types of factors: physiographic, socioeconomic, and village governance factors. Among the tested variables, proximity to suburban areas, the proportion of non-local permanent residents, annual per capita income, and village infrastructure conditions have significant positive effects on the rural housing rental rate, whereas distance from the nearest county government/district government, distance from the nearest township government/sub-district, reliance on funding from the higher-level authority of the village committee, and the degree of harmony between villagers and village committee cadres have significant negative effects. Their rate was not significantly associated with the proportion of residents working and doing business outside the village or with the number of village representative meetings/village assemblies held in the village.

## 7. Interpretation of Policy Implications

Based on the above research findings, the following policy implications can be drawn. First, rural housing rental behavior is not limited to economically developed provinces and cities on the eastern coast but also exists to varying degrees in the vast inland provinces and cities of central and western regions. Therefore, when carrying out the top-level design of homestead system reform, it is necessary to fully consider the policy demands of central and western rural areas to realize the asset attribute of homesteads, giving rural residents throughout China the opportunity to increase their property income.

Second, there are currently wide regional differences in the rural housing rental rate, and the foundation and conditions for promoting reform of the homestead system vary between regions: in some provinces and municipalities, the rural housing rental market has developed to a certain scale; in others, that market is just beginning to develop. Consideration can be given to early and pilot implementation in some provinces and municipalities with relatively good conditions, so as to promote the reform of separating rural land ownership rights, contract rights, and management rights on a larger regional scale and at a higher level of policy coordination. This exploratory approach would help to accumulate experience and provide typical demonstrations for other Chinese regions.

Third, the development of a non-farm economy in rural areas, the inflow of non-local population, and the improvement of rural infrastructure are important driving forces for developing the rural housing rental market. With the continuous promotion of the rural revitalization strategy and the two-way flow of population between urban and rural areas, it is necessary to pay greater attention to the immediate housing needs of rural migrant workers and businesspeople, while also meeting the urgent demand of farmers to increase their property income. To this end, restrictions on the right to use homesteads should be moderately relaxed.

Fourth, in rural areas far from towns, there are vast areas that the urban housing rental market cannot cover. Fostering the rural housing rental market and promoting the construction of relevant support systems can effectively meet the real housing rental demand in remote rural areas and achieve functional complementarity with the urban housing rental market.

Fifth, as the owners of homesteads, village collective organizations are both the main body for formulating and implementing homestead management policy and the specific field for relevant stakeholders to play benefit games and interact with each other. Improving the democratization degree of village decision-making, creating harmonious relations between cadres and the masses, and enhancing the governance level of village organizations could all significantly foster the rural housing rental market, thereby increasing farmers' property income and effectively promoting the reform of separating rural land ownership rights, contract rights, and management rights of the homestead.

Although the debate on rural land issues in China is intense, there is a basic consensus that the continuous advancement of industrialization and urbanization have brought

profound changes to the homogeneous small farmers of traditional rural China who live off the land, and that farmers are now highly differentiated in their involvement in non-farm economic activities [59]. It is now difficult to discuss peasants' rights, including land rights, in abstract terms using generalized conceptions of peasant, rural area, or peasant land [60], and it is necessary to discuss specifically and purposefully the land rights and rural area in question [61]. Based on this basic consensus, this paper used CLDS village residence data to empirically analyze regional differences in the rural housing rental rate and their influencing factors, aiming to provide a reference for locally appropriate and categorical reform of the homestead system. To deepen understanding of changes in the functional attributes of homesteads, future research should use household survey data collected from a larger sample, investigate more factors, and analyze the functional coordination between the residential security and asset attributes of homesteads. At the same time, we need to emphasize the policy implications in this paper cannot provide a direct-action basis and a ready-made action plan for the rural land system reform in developing countries due to different basic land systems. However, different countries try to develop non-agricultural economies, attract a floating population, improve rural infrastructure, and enhance village governance levels by virtue of their natural geographical conditions, which yields universal guiding significance and policy reference value for cultivating rural land markets and further increasing farmers' property incomes.

**Author Contributions:** Conceptualization, L.H. and R.W.; methodology, L.H.; software, M.Z.; validation, L.H.; formal analysis, L.H.; investigation, R.W.; resources, L.H.; data curation, L.H. and M.Z.; writing—original draft preparation, L.H. and R.W.; writing—review and editing, L.H.; visualization, L.H.; supervision, R.W.; project administration, L.H. and R.W.; funding acquisition, L.H. and R.W. All authors have read and agreed to the published version of the manuscript.

**Funding:** This research received funding from the National Natural Science Foundation of China (Project Number: 72004211, 72103175) and the Research Foundation of the Ministry of Education of P.R.China (Project Number: 21YJCZH164, 20JZD013).

**Institutional Review Board Statement:** Not applicable.

**Informed Consent Statement:** Not applicable.

**Data Availability Statement:** The data that support the findings of this study are openly available from China Labor-force Dynamic Survey (CLDS) at http://css.sysu.edu.cn (accessed on 6 July 2022).

**Acknowledgments:** We would like to thank Jian Tang and Rong Tan for their helpful comments on earlier drafts. The valuable reports from two anonymous reviewers are also much appreciated.

**Conflicts of Interest:** The authors declare no conflict of interest.

## Notes

1. The Ministry of Land and Resources was restructured into the Ministry of Natural Resources under the new round of administrative system reforms in China that began in 2018.

2. Since the founding of the PRC, the provision of rural social security has remained at a low level, and the planned allocation system has somewhat compensated peasants' loss of homesteads through collective membership rights, which gives farmers a kind of housing security.

3. Due to the impact of survey costs, the 2012 national baseline survey (CLDS2012) did not include the provinces of Xinjiang, Qinhai, Tibet, Hainan, and Taiwan. The first follow-up survey (CLDS2014) and the second follow-up survey (CLDS2016) did not include the provinces of Qinhai, Tibet, Hainan, and Taiwan.

4. As of 6 July 2022, the third follow-up survey (CLDS2018) is the latest data released by the China Labor-force Dynamic Survey (CLDS). However, the authors of this paper have not been permitted to use CLDS2018 data for the time being. We will update the empirical results and research findings according to the release of CLDS data in the follow-up study in due time.

5. According to the division of the four major economic regions proposed by the National Bureau of Statistics of China, east China includes Beijing, Tianjin, Hebei, Shanghai, Jiangsu, Zhejiang, Fujian, Shandong, Guangdong, and Hainan; mid-China includes Shanxi, Anhui, Jiangxi, Henan, Hubei, and Hunan; west China includes Inner Mongolia, Guangxi, Chongqing, Sichuan, Guizhou, Yunnan, Tibet, Shaanxi, Gansu, Qinghai, Ningxia, and Xinjiang; and northeast China includes Liaoning, Jilin, and Heilongjiang.

[6]    The lack of data about provinces such as Tibet, Qinghai, and Hainan results from the fact that the China Labor-force Dynamic Survey has not been carried out in the above provinces because of insufficient investigation costs. Meanwhile, the lack of data about the above-mentioned provinces may lead to the overestimation of the rural housing rental rate throughout China, the eastern region, and the western region to some extent.

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
