# Peer review of "Rural Housing Rental Rates in China: Regional Differences, Influencing Factors, and Policy Implications"

_land, doi:10.3390/land11071053_

Round 1

Reviewer 1 Report

Dear Authors,

Thank you for your work in this area,

This manuscript has research value and innovation.

But there are still many adjustments, not many, but essential.

The followings are my comments for this current version.

1. The current format still keeps the previous submitted journal's format. Please apply the Land's template strictly, for instance, affiliation, reference citation, line number, etc. Now there are many format errors, making it difficult to review the manuscript.

ref:

[1] https://www.mdpi.com/files/word-templates/land-template.dot

2. Without the line number, there is no way to locate the comment in your manuscript. I pointed out the comment. You need to find the specific location in the manuscript and modify it.

3. Chapter 1, 'slum' is a definition in a capitalist country. In China, a socialist country, there is no slum. If it is used persistently, it may cause unnecessary political problems. Because this means the failure of the Communist Party's urban construction management. Please check that your references match the manuscript.

ref: 

[1] http://politics.people.com.cn/n1/2019/1118/c429373-31460835.html

4. The international value of this manuscript is limited because you have mentioned many times that China's land, whether in urban or rural areas, is owned by collectives or governments, and private people only have the right to use it, which is different from most mainstream countries in the world. Please adjust the research significance of the corresponding part of the manuscript. You need to point out the reasons for the absence of Tibet and Hainan, and explain the impact of this problem on your final results and manuscript.

5. Section 3.1. Why does your database not include Tibet and Hainan? you should explain further, because the research range of your title, is China, but your research data does not cover all regions of China. Not including Taiwan, is ok, because it's the republic of China, not the people's republic of China. They are two different countries. But why no Tibet and Hainan?

6. Figure 2 is too general, your manuscript is published in an international journal, and the readers and reviewers are from worldwide. not all of them are familiar with the administration distribution of Chinese regions. You need to point out the province name corresponding to each map block on the geographic map, otherwise, this map is meaningless to most people. Because they don't know where each block corresponds, and they can't correspond to your influencing factors chapter.

ref:

[1] https://www.researchgate.net/figure/Spatial-distribution-of-the-U5MR-in-China-from-2001-to-2010-a-j_fig1_321208252

7. To be honest now is 2022, and your database is a little bit delayed, the last update is 2016, six years ago. especially with the outbreak of the covid-19, the details of your research object must be influenced.

8. Doing research on spatial distribution characteristics usually applies ArcGIS methods, like, kernel density and imbalance index, but considering your research field, The defects in this part are acceptable. But at least, you should explain clearly the spatial distribution situation, in Table 2 and Figure 2, they are distributed at the administration provinces level. but others, like in Table 3, you are presenting them in geographical division distribution, at least, you should explain the relationship between these two levels. Which province corresponds to which division and which division contains which provinces. Besides, you also have problems with the standard of geographical division. Your division does not include the south.

ref:

[1] Spatial distribution of China׳s renewable energy industry: Regional features and implications for a harmonious development future.

[2] Spatial Distribution Characteristics and Influencing Factors of Traditional Villages in China

[3] Analysis of spatial structure and influencing factors of the distribution

9. China's rural land problem is very complex, and the manuscript does not elaborate in detail. For example, in many urban villages in Shenzhen and Guangzhou, there are a large number of self-built houses and small property rights houses for rent, but they all belong to rural land rent. These issues are generally discussed in the manuscript. if deepen the analysis and consideration of this part, which will affect the current research results, and improve the depth and breadth of the article and the significance of the research to a great extent.

10. I strongly suggest that you divide Chapter 6 into two chapters, 6. discussion and 7. conclusion.

Author Response

Reply report to reviewer 1#

(Reply is written in dark blue)

Comments and Suggestions for Authors

a.The current format still keeps the previous submitted journal's format. Please apply the Land's template strictly, for instance, affiliation, reference citation, line number, etc. Now there are many format errors, making it difficult to review the manuscript.

Reply: Thank you very much. We have corrected the format errors in the manuscript by strictly following Land's template. Meanwhile, we use the English editing service provided by MDPI to correct the format errors in the manuscript.

b.Without the line number, there is no way to locate the comment in your manuscript. I pointed out the comment. You need to find the specific location in the manuscript and modify it.

Reply: Thank you. We have added the line number to the manuscript. Besides, the specific line number of each modification was noted in the process of revising the manuscript.

c. Chapter 1, 'slum' is a definition in a capitalist country. In China, a socialist country, there is no slum. If it is used persistently, it may cause unnecessary political problems. Because this means the failure of the Communist Party's urban construction management. Please check that your references match the manuscript.

Reply: Thank you. We have replaced "slum" with "shanty town" at Line 78.

d.The international value of this manuscript is limited because you have mentioned many times that China's land, whether in urban or rural areas, is owned by collectives or governments, and private people only have the right to use it, which is different from most mainstream countries in the world. Please adjust the research significance of the corresponding part of the manuscript. You need to point out the reasons for the absence of Tibet and Hainan, and explain the impact of this problem on your final results and manuscript.

Reply: Thank you. As you said, China has a special land system in contrast with most developing countries. In particular, the land ownership is owned by the Chinese government or collectives. We adjusted the description of research signification at line 716-723. It was clearly pointed out that the policy implications in this paper cannot provide a direct action basis and a ready-made action plan for the rural land system reform in developing countries due to different basic land systems. However, different countries try to develop non-agricultural economy, attract floating population, improve rural infrastructure, and enhance village governance level by virtue of their natural geographical conditions, which yields universal guiding significance and policy reference value for cultivating rural land markets and further increasing farmers' property incomes.

At the same time, Note 6 was added at line 368, which pointed out that the lack of data about provinces such as Tibet, Qinghai, and Hainan results from the fact that the China Labor-force Dynamic Survey hasn’t been carried out in the above provinces because of insufficient investigation costs. Meanwhile, the lack of data about the above mentioned provinces may lead to the overestimation of the rural housing rental rate throughout China, the eastern region, and the western region to some extent.

e.Section 3.1. Why does your database not include Tibet and Hainan? you should explain further, because the research range of your title, is China, but your research data does not cover all regions of China. Not including Taiwan, is ok, because it's the republic of China, not the people's republic of China. They are two different countries. But why no Tibet and Hainan?

Reply: Thank you. The China Labor-force Dynamic Survey was not carried out in all provincial administrative regions in China owing to the influences of investigation costs, etc. The 2012 national baseline survey (CLDS2012) didn’t include the provinces of Xinjiang, Qinhai, Tibet, Hainan and Taiwan. The first follow-up survey (CLDS2014) and the second follow-up survey (CLDS2016) didn’t include the provinces of Qinhai, Tibet, Hainan and Taiwan. Note 3 was added at line 243, explaining the aforementioned problem we encountered.

.

f. Figure 2 is too general, your manuscript is published in an international journal, and the readers and reviewers are from worldwide. not all of them are familiar with the administration distribution of Chinese regions. You need to point out the province name corresponding to each map block on the geographic map, otherwise, this map is meaningless to most people. Because they don't know where each block corresponds, and they can't correspond to your influencing factors chapter.

Reply: Thank you. We ignored our readers when making Figure 2. Thank you for pointing out this mistake. In the revised manuscript, we remade Figure 2, pointing out the province name corresponding to each map block on the geographic map.

g.To be honest now is 2022, and your database is a little bit delayed, the last update is 2016, six years ago. especially with the outbreak of the covid-19, the details of your research object must be influenced. 

Reply: Thank you. The third follow-up survey (CLDS2018) is the latest data released by China Labor-force Dynamic Survey (CLDS). However, we have not been permitted to use CLDS2018 data by the submission date (July 6, 2022) of this revised manuscript. Note 4 was added at line 257, explaining the aforementioned problem we encountered. In Chapter 5, we additionally introduced the related cases of “Activation Plan of Idle Rural Houses” of Shaoxing, Zhejiang Province (line 555-563)and the autonomous management mode of homestead of Yujiang District, Jiangxi Province(line 574-583), revealing the flexible policy adjustment by the Chinese local governments to cultivate rural land markets and increase farmers’ property incomes after the COVID-19 pandemic broke out.

h. Doing research on spatial distribution characteristics usually applies ArcGIS methods, like, kernel density and imbalance index, but considering your research field, The defects in this part are acceptable. But at least, you should explain clearly the spatial distribution situation, in Table 2 and Figure 2, they are distributed at the administration provinces level. but others, like in Table 3, you are presenting them in geographical division distribution, at least, you should explain the relationship between these two levels. Which province corresponds to which division and which division contains which provinces. Besides, you also have problems with the standard of geographical division. Your division does not include the south.

Reply: Thank you. According to the division of the four major economic regions proposed by the National Bureau of Statistics of China, east China includes Beijing, Tianjin, Hebei, Shanghai, Jiangsu, Zhejiang, Fujian, Shandong, Guangdong, and Hainan, mid-China includes Shanxi, Anhui, Jiangxi, Henan, Hubei, and Hunan, west China includes Inner Mongolia, Guangxi, Chongqing, Sichuan, Guizhou, Yunnan, Tibet, Shaanxi, Gansu, Qinghai, Ningxia, and Xinjiang, and northeast China includes Liaoning, Jilin, and Heilongjiang. We added Note 5 at line 264, introducing the division of China’s four major economic regions to readers.

i.China's rural land problem is very complex, and the manuscript does not elaborate in detail. For example, in many urban villages in Shenzhen and Guangzhou, there are a large number of self-built houses and small property rights houses for rent, but they all belong to rural land rent. These issues are generally discussed in the manuscript. if deepen the analysis and consideration of this part, which will affect the current research results, and improve the depth and breadth of the article and the significance of the research to a great extent. 

Reply: Thank you very much for the valuable suggestion. At line 613-627, we added three typical “villages in cities” cases in Yuancun Village, Hecun Village, and Fangcun Village in the Pearl River Delta region, revealing the influences of “degree of harmony between villages and village committee cadres” on rural housing rental rate.

j. I strongly suggest that you divide Chapter 6 into two chapters, 6. discussion and 7. conclusion.

Reply: Thank you very much. According to your suggestion, we have divided Chapter 6 into two chapters, 6. research findings and 7. interpretation of policy implications.

Reviewer 2 Report

As the title suggests, this paper reports an analysis of rural housing rental rates in China. The introduction and the second section contain an excellent blend of history and literature review of rural land and rural housing policy in China. Some readers may be familiar with parts of this complex and fascinating story but the authors have done a fine job of unifying the story and setting the institutional context for the analysis that follows.

The data set used is from the China labor force Dynamics Survey (CLDS).  This is a large, well-designed survey conducted every two years by Sun Yat-sen University.  The data set is certainly appropriate to the task at hand.

The methodology consists of three main approaches: a decomposition of Dagum Gini coefficients, a logit model and a Tobit model. All three methods are explained clearly and thoroughly, as are the justifications for using each method.  The methods are appropriate for the task at hand.

The results are explained clearly, and readers should have little difficulty understanding the results or the well-discussed policy implications, which follow directly from the results.

Two minor comments:

    In the introduction (last paragraph before section 2, is the following statement “The second section develops a theoretical framework …” Section 2 is not really a theoretical framework. Section 2 sets the institutional context for the study. So this statement should be changed to reflect the contents of section 2.

    The CLDS data are from the 2012, 2014, and 2016 surveys. No mention is made of later surveys even though the surveys are conducted every two years. Why? There may be good reason not to use data from the later surveys or perhaps the later surveys don’t exist?

Author Response

Reply report to reviewer 2#

(Reply is written in dark blue)

Comments and Suggestions for Authors

a.In the introduction (last paragraph before section 2, is the following statement “The second section develops a theoretical framework …” Section 2 is not really a theoretical framework. Section 2 sets the institutional context for the study. So this statement should be changed to reflect the contents of section 2.

Reply: Thank you very much for the valuable comment. We have replaced the statement “The second section developments a theoretical framework …” with “The second section details the institutional background of the study” at line 110-111.

b.The CLDS data are from the 2012, 2014, and 2016 surveys. No mention is made of later surveys even though the surveys are conducted every two years. Why? There may be good reason not to use data from the later surveys or perhaps the later surveys don’t exist?

Reply: Thank you very much. China Labor-force Dynamic Survey (CLDS) is carried out every two years, and its latest data is CLDS2018. We have tried many ways to contact the Institute of Social Science Survey of Sun Yat-sen University for the CLDS2018 data since the data is not accessible to the public. However, we have not been permitted to use CLDS2018 data by the submission date (July 6, 2022) of this revised manuscript. Note 4 was added at line 257, explaining the aforementioned problem we encountered. In Chapter 5, we additionally introduced the related cases of “Activation Plan of Idle Rural Houses” of Shaoxing, Zhejiang Province (line 555-563) and the autonomous management mode of homestead of Yujiang District, Jiangxi Province (line 574-583) , revealing the flexible policy adjustment by the Chinese local governments to cultivate rural land markets and increase farmers’ property incomes after the COVID-19 pandemic broke out for the purpose of making up for the outdated research data.

Round 2

Reviewer 1 Report

Thank you for your revision.